# Exosomes and Micro-RNAs in Aging Process

**DOI:** 10.3390/biomedicines9080968

**Published:** 2021-08-06

**Authors:** Yousra Hamdan, Loubna Mazini, Gabriel Malka

**Affiliations:** Institute of Biological Sciences, Université Mohammed VI Polytechnique, Lot 660 Hay Moulay Rachid, Ben Guerir 3150, Morocco; yousra.hamdan@um6p.ma (Y.H.); gabriel.malka@um6p.ma (G.M.)

**Keywords:** exosomes, aging, micro-RNA, senescence, longevity, molecular regulation, age-related disease

## Abstract

Exosomes are the main actors of intercellular communications and have gained great interest in the new cell-free regenerative medicine. These nanoparticles are secreted by almost all cell types and contain lipids, cytokines, growth factors, messenger RNA, and different non-coding RNA, especially micro-RNAs (mi-RNAs). Exosomes’ cargo is released in the neighboring microenvironment but is also expected to act on distant tissues or organs. Different biological processes such as cell development, growth and repair, senescence, migration, immunomodulation, and aging, among others, are mediated by exosomes and principally exosome-derived mi-RNAs. Moreover, their therapeutic potential has been proved and reinforced by their use as biomarkers for disease diagnostics and progression. Evidence has increasingly shown that exosome-derived mi-RNAs are key regulators of age-related diseases, and their involvement in longevity is becoming a promising issue. For instance, mi-RNAs such as mi-RNA-21, mi-RNA-29, and mi-RNA-34 modulate tissue functionality and regeneration by targeting different tissues and involving different pathways but might also interfere with long life expectancy. Human mi-RNAs profiling is effectively related to the biological fluids that are reported differently between young and old individuals. However, their underlying mechanisms modulating cell senescence and aging are still not fully understood, and little was reported on the involvement of mi-RNAs in cell or tissue longevity. In this review, we summarize exosome biogenesis and mi-RNA synthesis and loading mechanism into exosomes’ cargo. Additionally, we highlight the molecular mechanisms of exosomes and exosome-derived mi-RNA regulation in the different aging processes.

## 1. Introduction

Almost every cell, including stem cells, naturally release extracellular vesicles (EVs) responsible for cell-to-cell communication. Based on the evolution of the collective knowledge for the last decade, the International Society for Extracellular Vesicles endorsed EV as the generic term for particles deprived of the nucleus and delimited by a lipid bilayer without replication capacity [1]. EVs are differently sized vesicles released in vivo and in vitro and are classified based on the mode of release, size, and method of purification [1]. They are split into three categories: microvesicles or ectosomes are submicron vesicles with a diameter of 100 nm–1000 nm, distinguished by biogenesis mechanisms including cytoskeleton remodeling and phosphatidylserine externalization. These microvesicles are formed by the outward budding and fission of the plasma membrane (PM) after cell stimulation or stress [2,3,4,5,6]. The other EVs are apoptotic bodies with a diameter between 50 nm–4000 nm and are usually released during the stage of cell apoptosis [7]. Compared to the microvesicles, the apoptotic bodies contain contact organelles, chromatin, histones, and glycosylated proteins [3]. Exosomes are the most common EVs studied and the smallest ones with a diameter of 30 nm–100 nm. They were first considered as a cell metabolic product by the inward blebbing [6] and described as a part of disposal mechanisms discarding unwanted material from cells [8,9]. Their biogenesis is orchestrated from PM followed by the maturation process determining their cargo composition [7,10,11].

Exosomes are considered key regulators of many biological settings and are present in several extracellular fluids to mediate cellular communication. Recently, they were suggested as biomarkers for several diseases to set up diagnosis and disease progression. Their characteristics hold a great interest in designing therapeutic purposes in metabolic and genetic disorders, neurodegenerative and cardiovascular diseases, and cancer [12,13,14,15]. Nevertheless, different characterizations and validation methods have been developed whereby exosome purification and drug delivery are improved [1,15,16,17]. A recent study used a high-efficiency 3D nanostructured microfluidic ship to capture intact exosomes and a dual ligand to encapsulate the drug as an innovative and promoting methodology compared to ultracentrifugation and ultrafiltration [18]. The same team has also developed an engineered biological and synthetic hybrid designer exosome with photoresponsive functionalities targeting cancer cells [19].

Advancements in exosome identification and analysis have concluded on their enrichment of micro-RNAs (mi-RNAs), proteins, lipids, and DNA [1,14,20]. Mi-RNAs are described as a ~22-nt RNA, binding to their targets messenger RNAs (mRNAs) by base-pairing between 5′ end sequences of mi-RNA and mRNAs sequences located at 3′ untranslated regions (3′ UTR) [21]. Consequently, gene expression levels were found to contribute to fine-tuning and buffering the activity of different signaling pathways [22]. Increasing evidence has involved mi-RNAs in the regulation of different biological processes such as immune responses, apoptosis, metabolism, cancer, and cell aging [23,24,25,26,27,28]. These mi-RNAs are expressed differently according to the age and age-related diseases in blood [29,30,31], plasma [32,33,34], and serum [35]. Their expression was associated with aging mechanisms and targeted the transcription and translation of the molecular pathways involved in human longevity [32,34,36,37].

Epigenetic factors might also be age predictors through mi-RNAs expression levels [37]. Recently, different authors have proposed the involvement of lin-4, let-7, mi-RNA-17, and mi-RNA-34 as age and longevity-related mi-RNAs [36]. Other mi-RNAs were reported significantly different in long- versus short-lived individuals [36,38]. Most of these mi-RNAs were downregulated and predicted to target hundreds of genes outlined in many mi-RNAs databases making it difficult to understand the physiological pathways involved in longevity.

On the other side, the first signature of human aging is the decrease of tissue regeneration and repair, leading to the accumulation of senescent cells. These cells have been described to release more exosomes with different compositions than a normal cell [39]. Moreover, the senescence-associated secretory phenotype (SASP) was described to induce pro-inflammatory cytokines, chemokines, and tissue-damaging proteases [40]. Senescent cells were shown to accumulate senescence-associated ß-galactosidase (SA β-gal) activity and the senescence marker p16^Ink4a^ [41,42]. In vitro, early expanded mesenchymal stem cells (MSCs) also presented a stemness profile, while the long-term expanded were associated with a senescence phenotype [43,44]. A cellular transcriptional program is thus induced whereby the number and composition of exosomes are changed, consequently reflecting the current parent cell profile [45,46,47].

Much evidence has increasingly involved exosomes and exosome-derived mi-RNAs in both normal and pathological aging processes [48,49,50,51,52,53]. Evidence has also increasingly involved exosome-derived mi-RNAs in aging-associated diseases. In this work, we review exosome biogenesis and its involvement in the mechanisms related to aging with a focus on the different pathways described for their secreted mi-RNAs.

## 2. The INS and Outs of Exosomes

### 2.1. The INS of Exosomes: The Biogenesis

Exosome biogenesis remains incompletely understood even though different strategies have been described, especially the involved pathways. Biogenesis starts with the selective removal of the PM by an inward budding into the cell, constituting the intraluminal vesicles (ILVs). These endocytic vesicles mainly emerge from the lipid raft domains of PM and occur quite fast, so that 50–180% PM of reticulocyte can be recycled in only one hour [54], resulting in the intracellular generation of early endosomes [3,4,54,55]. Thereafter, these early endosomes can have different destinations depending on the enclosed cargo. The PM can be directly recycled by endosomes, and the MVBs can fuse with the lysosome if the content is targeted for degradation [3,55,56]. Early endosomes might mature to the late endosome by ILVs accumulation to multivesicular bodies (MVBs) [3,57]. Finally, these MVBs might fuse with the cellular membrane to release the ILVs as exosomes into the extracellular space [2,3,54,55,57] (Figure 1).

ILV formation or exosome biogenesis can be performed by two distinct processes. Both mechanisms occur depending on the physiological state and the type of the parent cell and are involved synergistically but might not be separated [58]. ESCRT proteins consist of four different protein complexes ESCRT-0, ESCRT-I, ESCRT-II, and ESCRT-III [3,11,59]. One of the most complete studies about ESCRT proteins was made by Colombo et al. in 2013, using an RNA interference screen targeting 23 ESCRT and their associated proteins VP4, VTA1, and ALIX [59]. Depleting ESCRT-0 and ESCRT-I and its associated proteins “Hrs, TSG101, and STAM-1,” respectively, reduced exosome secretion. Nonetheless, exosome oversecretion was reported as one of ESCRT-III, and its associated protein ALIX, CHMP-4C, VPS4B, and VTA1 were knockdowns, suggesting the involvement of the VPS4B protein [59]. In addition, silencing ALIX protein changed the exosome protein content rather than secretion, indicating its contribution to the exosomes cargo process [60]. Sterzenbach et al. used the L-domain containing protein Ndfip1 to increase ALIX protein recruitment in the ESCRT-pathway, thus loading exogenous proteins and delivering active proteins across the blood-brain barriers [61]. Moreover, depletion of Hrs, ESCRT-0 associated protein was associated with a decrease in exosome secretion and a reduced level of Wnr3A and Evi in human embryonic kidney 293 cells (HEK293) [62].

Tspans are other actors implicated in exosome biogenesis and can be used as exosome markers. The most-reported Tspans are CD9 and CD82 enhanced in the exosomal release of β-catenin from HEK293 cells. Knockdown of CD9 bone marrow dendritic cells secrete less affiliated flottin-1 suggesting the importance of this Tspan in exosome formation and release [63]. CD81 is involved in mRNA and protein composition in rat adenocarcinoma cell-derived exosomes [64]. A knockout of Tspan63 reduced exosome formation and secretion. A recent finding has demonstrated that the latent membrane protein-1 (LMP-1) is a major viral oncogene expressed in most Epstein-Barr viruses, suggesting a role in packaging and particle secretion by CD63 [65]. Actually, TSG101 and CD9, CD63, LMP-1, CD81, CD82 are used for exosome identification and characterization [1,60,66,67,68]. In a follow-up, the Syndecan-syntenin-ALIX pathway is a non-ESCRT-mechanism facilitating the ILV formation, exosome-containing syntenin, syndecan, and CD63 are released by heparanase and play a key role in endosomal membrane budding controlled by ARF-6 and PLD-2 [3,57,69]. This mechanism controls approximately 50% of the secreted exosome in the Michigan Cancer Foundation-7 (MCF-7) cells [70].

### 2.2. The Out of Exosomes: The Release

Several key molecules are involved in exosome secretion. The Rab families are small-molecule GTP-binding proteins known to play a pivotal role in transferring vesicles between intracellular compartments and trafficking MVB to the plasma membrane (Figure 1). Rab-5 and Rab-7 are common markers of early and late endosomes, respectively. In MCF-7 cells, Rab-7 was reported to regulate the secretion of syndecan and syntenin [69], resulting in the enrichment of exosomes with ALIX and syntenin [69]. Furthermore, blocking Rab-27 GTP prevented exosome secretion while silencing Rab-27-a and Rab-27-b resulted in the inhibition of docking with the cellular membrane redistribution to the perinuclear space [71]. Moreover, Rab-11 involvement is differently appreciated in exosome secretion. In the K562 human leukemic cell line, a recent study showed that this CD47 expression allows exosomes to escape from phagocytosis by macrophages and monocytes, consequently increasing their life shelf in blood circulation. However, CD47 does not play a significant role in exosome cell but enhanced micropinocytosis in Kras mutant cells and favored exosomes uptake [72]. Rab-11 also inhibits the exosomal release where no consequence was demonstrated in Henrietta Lacks (HeLa) cells [60].

Another point of view, the ionophore MON was shown to induce the formation of a large MVB and stimulates the release of exosome through a Ca^2+^ dependent manner in K562 cells. The same results were obtained by chloroquine and bafilomycin as they elevate the Ca^2+^ [73]. The dissociation of the subunit of V1-ATPase by autophagy protein 5 (Atg5) and autophagy protein 6L1 abolishes its proton transporter activity, and both reduce the acidification of the MVBs, increasing exosome release in human cells [74]. The Ca^2+^ dependent SNAP receptor Munc13-4 also regulates exosome release in metastatic cells [75]. The soluble N-ethylmaleimide-sensitive-fusion attachment protein receptor (SNARE) complex is another mechanism reported to facilitate the fusion of vesicles with their target membrane. In the K562 leukemic cell line, overexpression of VAMP-7, one of the SNARE proteins, reduced exosome release [76]. Another SNARE protein, YKT6, was related to exosome release in HEK293 human embryonic kidney cells and A549 human lung cells and involved in the secretion of Wnt-bearing exosome and sny-5 [62].

## 3. Composition of Exosome’s Cargo

Different technologies were used to characterize the exosome’s cargo, including transmission electron microscopy, scanning electron microscopy, atomic force microscopy, nanoparticle tracking analysis, dynamic light scattering, resistive pulse sensing, enzyme-linked immunosorbent assay, flow cytometry, fluorescence-activated cell sorting, microfluidics, and electrochemical biosensors [77]. Under electron microscopy, exosome morphology is cup-shaped after ultra-centrifugation and sucrose gradient step; they are highly enriched in CD63, ALIX, TSG-101, and HSC-10 [78]. Two sub-populations were described by Layden et al.: large exosomes with 90–120 nm average and small exosomes with 60–80 nm average [79].

Based on the type and size, exosomes cannot be distinguished from small microvesicles and other soluble molecules like exomeres [80]. Surprisingly, the same exosomes are expected to contain similar protein, nucleic acid, and lipid composition; however, their content might differ even from the same parental cell [81]. It was demonstrated that high cholesterol content is a maturation marker of MVBs and can participate in exosome release by fusing with plasma. However, exosomes secreted from the apical and basolateral sides of polarized cells are reportedly different in their composition, probably explaining the differences observed in MVB populations [60]. Moreover, exosome secretion levels can be increased, and their cargo can be enriched in specific proteins or mi-RNA by using genome editing [11,50,82,83].

Exosomes play an important role in the circulating secretome stability in blood and tissues; nonetheless, their composition remains dependent on their surrounding micro-environment [46]. On one side, Salomon et al. have published a large panel of proteins secreted by exosomes after MSCs exposition to different O_2_ concentrations. Different proteins such as Annexin A1, A2, and A3, ADP-ribosylation factor, BMP-1, COL6-A1, A2 and A3, BCL3, Heat Shock-90, PI3KR-4, TIMP-1, IL-25, actin, myosin, and integrin were differently quantified [47]. On the other side, exosome RNA levels reflect the cytoplasm composition [84]. The formation of these RNAs requires both nuclear and cytoplasmic phases leading to several non-coding RNA (long non-coding RNAs, mi-RNAs, circular RNA). Besides mi-RNA, other nucleic acids have been discovered in exosomes: PIWI-interacting RNA, small nucleolar RNAs, yRNAs, and vault RNAs [11,85]. Mi-RNAs are transcribed by RNA polymerase II and III, leading to the formation of pri-mi-RNA with a cap and a poly-A tail in the nucleus and a hairpin structure [86,87]. This pri-mi-RNA contains a terminal consisting of two flanking unstructured single-stranded tails and a double-stranded tail with about 30 base pairs [88]. Thenceforward, pri-mi-RNA is cleaved by a class 2 ribonuclease III enzyme (Drosha) and DiGeorge syndrome critical region 8 (DGCR-8), an RNA binding protein giving to a pre-mi-RNA of ~70 nt [87]. This pre-miR is transferred from the nucleus to the cytoplasm by exportin-5 [89] and further cleaved by an enzymatic complex set up by the Dicer endonuclease and TRBP or PACT protein activator of the interferon-induced protein kinase to deliver a mature duplex of mi-RNA of ~22 nt [90]. Thereafter, Dicer/TRBP assists the mature duplex mi-RNA complex loading into one of the four Argonaute proteins (AGO2), which are part of the RNA-induced silencing complex (RISC). Thus, heat shock protein-70 (Hsp-70) and heat shock protein-90 (Hsp-90) is also needed [91]. Once loaded, one of the mi-RNA strands, «the passenger strand» is expulsed, and the «guide strand» mi-RNA is retained in this complex to bind the functional mRNA target [12,92].

Moreover, various mechanisms were suggested describing mi-RNA loading into exosomes. Indeed, hnRNPA2B1, a ribonucleoprotein, has been reported to recruit mi-RNAs bearing exosomes by interacting with a four nucleotide motif (GGAG) [93,94]. The latter plays a crucial role in exosome formation but was not incorporated in it [94,95]. Mi-RNA is released through a ceramide-dependent secretory whose biosynthesis is regulated by neutral sphingomyelinase-2 (nSMase-2) [96]. Overexpression of this protein increased extracellular amounts of exosome-derived mi-RNA, as it is also described in exosome biogenesis [96]. The association of the AGO2 and the RISC complex are involved in RNA silencing and are believed to control the loading of mi-RNAs into exosomes [97]. Bone marrow mononuclear cells and myeloma cancer cell 9 showed that exosome contains mRNA and mi-RNA at a sufficient level to support complementary DNA synthesis even after treatment with RNase, suggesting the protective role of exosome [98].

## 4. General Molecular Mechanisms Related to Aging Dysfunction

Molecular science advancements have underlined the aging mechanisms and led to the identification of multiple key drivers. Cellular aging is due to various biological features and changes accelerating or delaying the longevity process. These hallmarks include genomic instability, telomere attrition, epigenetic alterations, loss of proteostasis, deregulated nutrient-sensing, mitochondrial dysfunction, cellular senescence, stem cell exhaustion, and altered intercellular communication [11,99]. Understanding these mechanisms might increase our knowledge of the multiple intracellular interactions governing cell aging and the balance between cell senescence and rejuvenation.

### 4.1. Genome Instability and DNA Damages

In normal conditions, cells do not divide indefinitely because of the termed process of replicative senescence as a permanent state of growth arrest. It was estimated that tens of thousands of damaging events occur each day in every single cell. The genotoxic attacks can originate from extrinsically inflicted radiation damage or chemicals as well as from endogenous sources [100]. One cannot define and explain aging without the cell changes during senescence induction.

DNA damages occur with age when single-stranded or double-stranded DNA breaks. These changes are mostly generated extrinsically by radiation or chemical toxicity. Two distinct outcomes are identified: cancer by erroneous repair leading to mutation and chromosomal aberration and cellular senescence by blocking transcription and replication to prevent tumorigenesis [100]. However, some studies have reported the crucial role of the mini-chromosome maintenance 2-7 (MCM2-7) [101,102]. The protein family is essential for DNA replication and binds to replication origins during the G_1_ phase of the cell cycle along with members of the pre-replication complex (origin recognition complex, cell division cycle 6, chromatin licensing, and DNA replication factor 1). Once DNA replication is initiated, the MCM2-7 proteins were shown to move along with the replication fork, further supporting their possible role as a replicative DNA helicase [103]. P53 is another protein considered as a tumor suppressor gene playing an important role in controlling cellular proliferation in the context of DNA damage. During genomic DNA duplication, p53 transcriptionally activates the mi-RNA-34, thus downregulating the MCM2-7 factor and resulting in cell cycle arrest and senescence [49,101]. Additionally, Zhang et al. found that mi-RNA-124 inhibited keratinocyte proliferation and collagen biosynthesis, while activation of Wnt/β-catenin occurred by targeting stress-associated endoplasmic reticulum protein-1 [104]. This mi-RNA-124 was also reported to inhibit proliferation, migration, and invasion of malignant melanoma cells (A375) [105]. Moreover, UVB irradiated normal human epidermal keratinocytes resulted in DNA damage and increased mi-RNA-124 expression after a comparison between young and old skin. Mi-RNA-124 was found to be upregulated in aged humans, thus inducing a senescent profile. Interestingly, a forced expression of mi-RNA-124 in squamous cell carcinoma resulted in a decrease in cell number [106].

### 4.2. Mitochondrial Decline

Mitochondrial activity is crucial for cell function and is implicated in a plethora of metabolic networks, and plays a significant role in sustaining the life and health of humans [107]. For instance, mitochondria do not display any repair mechanisms to eliminate DNA lesions; they reveal a base excision against oxidative damage, and the mutation frequencies in mtDNA are greater than nuclear DNA. When this happened, dysfunctional or damaged mitochondria are removed by autophagy and restored by fusion with healthy elements of the mitochondrial network. However, during aging, autophagy declines, and fission outpaces fusion leading to the accumulation of dysfunctional mitochondria [49]. The first evidence that mitochondrial DNA (mtDNA) might be important for aging and aging-related diseases was derived from the identification of human multisystem disorders, which were caused by mtDNA mutations that partially phenocopy aging. Besides, mtDNA doesn’t have histones, which increases its susceptibility to oxidative stress-induced damage [108]. Defective mitochondria will generate ROS, resulting in cellular aging and related disorders. Cells from mutant mice deficient in mtDNA polymerase showed impaired mitochondrial function but were not accompanied by an increase in reactive oxygen species (ROS) production [99]. One of the common features of aging is mitochondrial dysfunction associated with the decline of cell function and tissue decay [109].

Degenerated mitochondria are identified by PINK1, then E3 ligase parkin is recruited, and the channel protein VDAC1 is localized to the outer mitochondrial membrane. Finally, the autophagic adaptor p62/SQSTM1 recognizes and triggers mitophagy [110]. A recent study reported that the upregulation of mi-RNA-155 disturbed mitophagy in MSCs by targeting BCL2 associated athanogene 5 (BAG5), which directly interacts with PINK1 and protects mitochondria from degradation and mitophagy through the ubiquitin-proteasome system [111]. Therefore, mi-RNA-103a-3p was upregulated and target Parkin/Ambra-1 mRNA in the human brain and Parkinson’s disease leading to their decrease. A defect in the mitophagy process occurs, suggesting that mi-RNA-103a-3p inhibition shows protective effects in neuronal aging-related disease [112].

Moreover, the free-radical theory of aging implies that the production of ROS results in aging where the antioxidative system fails to remove oxidative damages such as DNA, protein, and lipids. Mi-RNA-335 and mi-RNA-34a can modulate mesangial cell senescence via inhibition of the expression and function of the mitochondrial antioxidative enzymes superoxide dismutase 2 (SOD2) and thioredoxin reductase 2 (Txnrd2), both suspected to play a key role in modulating cellular aging by detoxifying ROS generated in the mitochondria. Aging mesangial cells exhibited significant upregulation of mi-RNA-335 and mi-RNA-34a, inducing premature senescence with a concomitant increase in ROS [113]. In line with ROS production, Sirtuins (SIRT) were largely described for the regulation of lifespan and metabolism control due to the progress of mitochondrial function during dietary restriction. In mammals, seven members of the SIRT protein family known as class III histone deacetylase have been identified for their enzymatic activities, molecular functions, and involvement in diseases [114]. For instance, SIRT-1 was described to regulate longevity through maintenance of mitochondrial homeostasis [115] and may have a central role in mitophagy through regulation of mitochondrial uncoupling protein 2 (UCP-2) by targeting PPARG coactivator 1 alpha (PCG-1α). SIRT-1 inhibited mitophagy by depressing PCG-1α and UCP-2, and its inhibition may lead to the activation of DNA damage [116]. Lang et al. linked SIRT with cellular senescence and mitochondrial function. The upregulation of mi-RNA-15b negatively impacted stress-induced SIRT-4 expression countering senescence-associated mitochondrial dysfunction and possibly regulating organ aging [117]. SIRT-3 plays a key role in human physiology; its impact on nuclear and muscular function reduces the aging effect by its ability to regulate thermogenesis, DNA repair, and anti-oxidative stress. Knockout of SIRT-3 in mice led to mitochondrial dysfunction in the heart with cardiac hypertrophy [118]. Additionally, SIRT-3 interacts with the transcription factor forkhead box-O3-a and appears to regulate intracellular processes like cellular resistance to oxidative stress and general metabolism [119]. Interestingly, SIRT-7-catalysed H3K122 is significantly implicated in DNA-damage response and cell survival, suggesting its implication in cellular function during aging [120]. SIRT-6 is shown to play different roles in the metabolism of aging. The intensification of mitochondria stress signaling through the change of NAD^+^ levels is described as a target to improve mitochondrial function and prevent or treat age-associated decline [115].

### 4.3. Telomeres Involvement

Telomere constitutes a repetitive DNA sequence “TTAGGG” located at the end of chromosomes, playing a protective role from degradation. Human telomeres typically range between 10 to 15 kb. In 1961, Leonard Hayflick discovered the in vitro limit of somatic cell division called later the “Hayflick limit”. Telomeres were proposed as mitotic clocks for cell division [121]. Accordingly, human fibroblasts dividing in a cell culture died after nearly 50 divisions. The cells stopped dividing and underwent a programmed cell death [122]. Telomere shortening is explained by the fact that adult cells do no express DNA telomerase to replicate telomeres [49]; 30 to 200 sequences present at the end will be lost at every replication [123]. Additionally, telomeres are covered by a complex of six proteins called shelterin: TRF-1 determines the structure of telomeric end, TRF-2 implicated in telomerase protection and telomerase length homeostasis, TIN-2, RAP-1 POT-1, and TPP-1 [124].

When a telomere shortens to a certain length, the cell acquires the senescence status. This finding has been observed in several diseases such as cardiovascular diseases, cancer initiation, type 2 diabetes, and weakened immune function [99,125]. Nonetheless, mi-RNA has been described to regulate telomerase activity. Hrdličková et al. determined six mi-RNAs, mi-RNA-133a, mi-RNA-138-5p, mi-RNA-491-5p, mi-RNA-342-5p, and mi-RNA-541-3p regulating the telomerase catalytic subunit through TERT and involving the Wnt pathway [126]. Mi-RNA-138 has been reported to regulate cell longevity by controlling the telomerase reverse transcriptase, while mi-RNA-155 targets TRF-1 [49,127]. On the other side, mi-RNA-490 was reported to regulate the telomere maintenance program by directly targeting TRF-2 [128]. Besides, acute exercise can transcriptionally regulate several key telomeric genes. Four mi-RNAs were involved in telomeric gene mRNAs mi-RNA-15-a, mi-RNA-96, mi-RNA-181, and mi-RNA-186 after 60 min post-exercise [129]. The mi-RNA-34 family is related to telomere length and displays a regulatory role in stem cell division and cancers, and is actually proposed as the mi-RNA of longevity [36,125]. Recently, mi-RNA-185 was shown as a key aging-related biomarker accelerating the replicative senescence process in primary human fibroblasts in a POT-1 dependent manner [130].

### 4.4. Epigenetic Impact

Epigenetic means “outside conventional genetics” and is considered a key regulator of the aging process allowing genome changes without sustainable modification of the DNA sequence [49,99]. These modifications can be spontaneous or driven by external or internal factors. Those affecting longevity acted primarily via the modification of the epigenome, adding complexity to the aging process [131]. Different epigenetic mechanisms have been described, including DNA methylation [132,133,134], histone modification (including post-transcriptional modifications, structural and functional variants of the histones) [134,135], and transcription of noncoding mi-RNA [49,132,136].

The most described epigenetic mechanism is DNA methylation, where DNA is catalyzed by the DNA methyltransferase (DNMTs) family and transfers the methyl group from S-adenyl methionine to the fifth carbon of cytosine residue to form 5mC. This mechanism regulates gene expression by restraining or repressing the binding of a transcription factor by recruiting proteins [133]. Dnmt3a and Dnmt3b are described as a new methylation pattern adding the methylation to the unchanged DNA, while Dnmt1 functions during DNA replication and copies the existing parental DNA methylation [137]. DNMT1 plays a key role in maintaining genomic methylation patterns, and its activity seems to decrease during aging [138].

The methylation of mi-RNA-9-1 was found to increase the chance of contracting diabetic retinopathy disease by five-fold, and both methylations of mi-RNA-9-1 and mi-RNA-9-3 increased this chance by eight-fold [139]. Additionally, downregulation of mi-RNA-29b is promoted via Dnmt3b, and its overexpression is found to prevent osteoarthritis through the Dnmt3b/mi-RNA-29b/PTHLH/CDK4/RUNX2 axis, thus suppressing chondrocyte apoptosis and extracellular matrix degradation [140]. Furthermore, communication between mitochondria and the nucleus is a must for any cell, and the disruption of this mechanism is implicated in cancer development. Mi-RNA-663 is regulated epigenetically by mitochondria, and its downregulation by DNA methylation and was described to reduce patient survival [141].

Epigenetic modification has also opened the way to post-transcriptional changes restoring specific cell functions. Although, their use in therapeutic purposes raised hope in treating metabolism, degenerative, and even genetic diseases. Chronic myelomonocytic leukemia is one of the most aggressive cancers affecting hematopoietic stem and progenitor cells. Downregulation of mi-RNA-125a is mediated by hypermethylation of its upstream region and was used as leukemic treatment [142]. Also, increasing Wnt10-b was associated with the decrease of mi-RNA-7113 and the H3K4me3 methylation increase, which resulted in an inflammatory profile of immune cells [143].

## 5. Exosomes Implication in Aging Process

During aging, cells accumulate genetic damages, epigenetic changes, gene expression alterations leading to cancer, age-associated diseases, and neurodegenerative disorders [87,99,125]. Moreover, defects in molecular secretomes such as growth factors, chemokines, mRNA, mi-RNA, and signaling molecules could highly interfere with the aging machinery where exosomes could be the key. Adding to that, exosomes are responsible for intercellular communications mediating signal transmissions between surrounding and distant tissues in normal and pathological conditions [6,8,12]. These characteristics have opened the way to their use as biomarkers of tissue dysregulations and as intercellular drug delivery [50,55,56,63,70,76].

Since the last two decades, evidence has increasingly identified the pivotal role of exosomes in regulating aging mechanisms [52,53,144]. By targeting mRNA transcription and translation, the exosome ‘s cargo is presumably related to and impacted by the status of the parent cell and its microenvironment. Despite exosomes containing several proteins, lipids, and DNA, among other compounds, mi-RNAs have attracted more attention because of their implication and interconnection with biological mechanisms; their expression is considered the main reflection of the responses induced by exosomes [55,84,119,144]. It is largely accepted that exosomes act through their mi-RNAs in paracrine and autocrine pathways to modulate their micro-environment and induce cell senescence [28,36,41,107,145,146,147,148]. Moreover, exosome implication in the aging process is also closely related to their parent cells [47,149]. When released from senescent cells, SASP, including exosomes, spread senescence to the surrounding microenvironment and to young cells [42,145,146,150,151]. Additionally, senescent cells seemed to secrete more exosomes than the younger ones, especially in cancer conditions [152], while their human plasma concentration decreased during age in normal conditions [153]. However, exosomes were identified to promote cell migration, proliferation, and anti-senescence when released from stem cells [145]. Interestingly, their cargo was correlated to the proliferation and lifespan of fibroblasts [146]. In the same way, highly purified exosomes derived from induced pluripotent stem cells (iPSCs) reduced the level of ROS expression when added to senescent cells and resulted in an anti-aging phenotype [51]. Normal human dermal fibroblast aging and photoaging have also been alleviated phenotypically and genotypically after adding iPSCs conditioned medium containing exosomes [52,154].

Another point of view is that aging is associated with the reduction of the regenerative and reparative activity in organs and tissues, leading to the increase of senescent cells. Different findings have highlighted the interaction of aging with exosomes’ composition. Exosomes from young donors appeared more proliferative, promoting osteogenic differentiation and maturation of MSCs through the enriched protein Galectin-3 being decreased in plasma exosomes of aged individuals [155]. During aging, senescent cells are negatively impacted in their exosome composition of mi-RNAs and proteins, including transforming growth factors (TGF-β), growth differentiation factor-11 (GDF-11), interleukin-6 (IL-6), toll-like receptor 2 (TLR-2), TLR-4, tumor necrosis factor-α (TNF-α), insulin-like growth factor (IGF), hepatocytes growth factor (HGF), and IL-1 [53]. Some miRs mi-RNAs are upregulated or downregulated in senescent cells and are identified as targeting specific genes and pathways of senescence, such as p53-p21 and p-16p-RB [147,156,157,158,159].

Nevertheless, inflammatory cytokine secretion and release, including TNF-α, IL-6, and IL-1β, induced the secretion of larger and irregular-shaped exosomes from stem cells. Therefore, these signaling pathways are involved in the modulation of different gene expression and signaling molecules involved in the cell cycle arrest, including *MAPK*, *VEGF*, *Wnt*, *NF-kB*, *JNK*, *CD4/6*, *SIRT1*, *Cyclin D1*, *E2F*, *PI3K-AKT*, and *IL-6* [53,68,147,148,158,159,160,161]. These interaction mechanisms might explain the changes induced by inflammatory cytokines in biological aging, also called inflamm-aging [162]. Actually, MSCs implication in aging process remains indisputable; among their role in secreting the rejuvenating factor GDF-11 within exosomes, these cells might change the polarization of macrophages from M1 to M2 profile in response to exosome derived anti-inflammatory cytokines under stress conditions, thus improving tissue regenerative capacity [158,163,164]. Moreover, stimulation of MSCs by interferon-gamma (IFN-γ) and TNF-α increased exosomes secretion and inhibited CD4^+^ T-cell proliferation, therefore, inducing an anti-inflammatory effect largely accepted as the main actor in tissue regeneration and aging, especially in the skin [40,53].

## 6. The Importance of mi-RNA in Aging Function

Different aging pathways involving mi-RNAs have been described to influence lifespan [27,36]. Otherwise, the rate of aging and lifespan is regulated by several signaling pathways controlling metabolism, nutrient sensing, endocrine signaling, and stress resistance. The mi-RNA discovery enhanced a lot of understanding in intracellular communication, making a new area of research as a biological marker in human serum and other biological fluids such as saliva, plasma, and urine. Established in 2008 as a biomarker for cancer, they were since mentioned for numerous diseases [144]. Different mi-RNAs can take part in multifunctional mechanisms but using a similar pathway. Inversely, the same mi-RNA can be the key modification of numerous biological functions and display a specific role in several recipient cell networks [28]. Understanding the multiple interactions between mi-RNA secretion and release, tissue-specific function, and signaling mechanisms interconnection remains challenging and needs further investigation.

In the aging context, mi-RNAs represent adaptative mechanisms to maintain organismal homeostasis in response to different alterations. They are recognized to modify the translation of mRNAs in target cell unbalancing, thus cell development between senescence and tumorigenesis [165]. Moreover, mi-RNAs and exosome-derived mi-RNAs circulate freely in the body and targeted neighboring cells, different adjacent tissues, or distant organs. Several mi-RNAs have been reported in age-related diseases like cardiovascular and neurodegenerative diseases and cancer, but their involvement in the aging process still needs to be elucidated. Different profiled circulating mi-RNAs in serum and long- versus short-lived individual studies have identified differently expressed age-dependent mi-RNAs [31,38,166]. Mi-RNA-92a, mi-RNA-222, mi-RNA-375 were found upregulated while mi-RNA-29b, mi-RNA-106b, and mi-RNA-130b were downregulated. Other mi-RNA levels are significantly associated to longer lifespan individuals and found upregulated including mi-RNA-211-5p, mi-RNA-1225-3p, mi-RNA-5095, let-7a-5p, mi-RNA-30b-5p, mi-RNA-126-3p, and mi-RNA-210 [32,35].

Table 1 summarizes some of the mi-RNAs and related molecular signaling involved in aging. Mi-RNA-29 was decreased in human serum in long-lived individuals [38]. In the human brain, an extensive alteration of gene expression occurs during aging, and principally mi-RNA-29 was upregulated to regulate intracellular iron homeostasis to limit excessive iron exposure in neurons [167,168]. Therefore, the loss of mi-RNA accelerated the expression of aging phenotypes in both global gene expression and histological levels in short-lived turquoise killifish brains [169]. In another report, mi-RNA-29 was established as the first mammalian mi-RNA that is directly implicated in the lifetime tradeoff between lifespan and reproduction in mutant mice [168]. Interestingly, mi-RNA-29 was upregulated in aged rodent muscle after individual analysis and led to the senescence of muscle by activation of Wnt-3 and the suppression of the expression of signaling proteins (p85α, IGF-1, and B-myb), thus contributing to muscle atrophy [149]. Moreover, a protective role was described in cardiac aging through the TGF-β/Smad pathway. The activation of TGF-β induces the accumulation of mi-RNA-29a and mi-RNA-29c, directly suppressing Suv4-20h histone methyltransferases and reducing H4K20me3 compromising thus DNA damage repair and genome maintenance. In this regard, this endogenous mechanism aimed to accentuate the age-dependent cardiac damage like hypertrophy and fibrosis [159,169].

Bone aging naturally results in the apoptosis of osteocytes through Cx43 decrease. Indeed, Cx43 maintains osteocytes viability through the gap junction communication via mi-RNA-21 downstream regulation leading to PTEN activity inhibition and preserving the Akt pathway. In old osteocytes, reduced Cx43 levels resulted in apoptosis and liberation of RANKL and HMGB1 and finally resorption of osteoclasts signal associated bone surfaces [171]. Furthermore, in wound healing, mi-RNA-21 overexpression promoted keratinocyte migration and boosted re-epithelialization through TGF-β, while in endothelial cells, angiogenesis was inhibited via RhoB repression [172,173]. Induction of epithelial-mesenchymal transition was also attributed to this mi-RNA [182].

Nevertheless, mi-RNA-181 plays an identical role in skin and brain, inducing cell death and senescence through different pathways, including SIRT [174,175]. The SIRT pathways have a great ability to induce longevity and decrease the appearance of age-related diseases, especially in skeletal development and homeostasis [147,182]. This mi-RNA is found to be down-regulated in elderly humans and is mostly found associated with the inflammatory pathways associated with age-related diseases [183,184].

Upregulation of exosomal mi-RNA-34 was reported in aged mice and induce senescence phenotype and decrease of SIRT expression in primary bone marrow cells. Similarly, cognitive impairment and brain aging mediated through SIRT1/mTOR signaling pathways was also demonstrated [148]. Interestingly, in myocardial injury, which is an age-related disease, mi-RNA-34a plays a protective role by reducing cell death and functional decline and inhibiting autophagy after ischemia-reperfusion by targeting TNF-α [176]. In humans, serum mi-RNAs profiling has demonstrated that mi-RNA-34 promotes lifespan prolongation [36]. These findings suggested that microenvironment and epigenetic factors are of great interest in defining the functional effect of mi-RNA.

Thenceforward, multiple studies have described different mi-RNAs playing a protective role in vascular calcification. Mi-RNA-204/mi-RNA-211 attenuated vascular calcification and induced aging in a paracrine way through BMP2 [177]. On the contrary, bone marrow stromal cells expressing mi-RNA-183-5p showed a decreased ability of Hmox1 to protect from oxidative stress and excessive inflammatory reactions resulting in senescence and a reduced osteogenic differentiation [185].

In epithelial cells, inhibiting miR-570-3p associated with oxidative stress by p38 MAP kinase-c-Jun restores the rejuvenating effect of SIRT, and therefore reduces the abnormalities related to cellular senescence [158]. Mi-RNA-135a-5ip inhibition was described to stimulate neural cell proliferation [179], suggesting that both mi-RNA-570-3p and mi-RNA-135a-5p might play a key role in aging regulation and considered as aging markers.

In the skin, both mi-RNA-127-3p and mi-RNA-124 were reported to induce dermo-fibroblast senescence by remodeling the extracellular matrix leading to age-related defects [106,180]. However, mi-RNA-124 promotes regeneration in rat liver after partial hepatectomy [183], indicating once again a tissue-dependent functionality.

## 7. Discussion

The aging process affects the whole body, including organs and biological systems, with a decrease in their reparative and regenerative potential and distinguishable and/or visible changes. Albeit different processes have already provided evidence in regulating the physio pathological aspect of aging, exosomes and mi-RNAs have progressively been associated with these age-associated diseases and also to lifespan expectancy and cell senescence [27,32,37,45,51,117,186,187]. In aged cells, DNA damages would lead either to apoptosis or senescence, the first one would result in complete cell removal, and surprisingly, increasing apoptosis fastens the aging process [188]. Senescence is rather characterized by an irreversible growth arrest and a combination of changes in cell morphology, function, and behavior [8]. According to Kowald et al., three theoretical scenarios related to senescent cell accumulation were considered: (i) the immune system declines with time due to the aging process, (ii) the immune system cannot recognize all senescent cells, and (iii) the SASP converts healthy cells into senescent cells [51,188]. A recent finding has demonstrated that avoiding SASP in long-term culture expansion resulted in less senescence-associated exosomes, thus being able to impact genetic information and immunoregulatory potential of the micro-environment [2]. Therefore, targeting senescence-associated exosomes would be relevant for clinical application.

The introduction of exosomes as biologically active carriers and as specific biomarkers for life expectancy and age-associated diseases is likely due to the protection of their cargo from degradation and the immune system [1]. The use of the ESCRT and non-ESCRT pathways to enrich this cargo is becoming a step forward for personalized medicine, especially as biologically active carriers to deliver specific biomolecules [189].

Intracellular communication requires exosome-based mi-RNAs production and exosome-mediated uptake. Mi-RNAs use the RISC complex on their mRNA targets for translational repression or degradation. There are currently 7068 identified mature human mi-RNA sequences, and each of them can target 3.5 million genes [36,190]. Among all the theories describing aging and health deterioration, we point out homeostasis biology at genetic, molecular, and organismal levels [189]. Despite the growing findings arguing the therapeutic benefit of exosome and exosome-derived mi-RNA, their key mechanism involved in aging remains to be elucidated. Indeed exosome enriched fractions are recognized as active pharmaceutical components of MSC-based therapies with increased scalability of manufacturing [191]. Also, the model of isolation have been developed lately [17] and several research findings have been describing the different way to increase the release and loading protein into exosomes [61].

Actually, young and senescent cells differ in their exosomes’ secretion levels, but also in their mi-RNAs secretion panel identified in biological fluids, which has a biological significance in metabolism, glycolysis, renal function, immunocompetence, tumor evolution, and brain memory function [8]. Recent findings have identified a different mi-RNA secretion profiling related to individual aging and the selected biological fluid [36,38,184]. The detection of mi-RNAs involved in longevity will provide important insights into the molecular basis of aging, low expression of mi-RNA-34, mi-RNA-425-5p, mi-RNA-21-5p, and mi-RNA-212-3p speculating that centenarians have survived diseases and cancer due to their lifestyle but also to a low inflammation level [192].

Some reports have also indicated that mi-RNA changes depend on the tissue and global mi-RNA across all cell types [36,193]. Most of these circulating mi-RNAs are reportedly downregulated probably to restore the altered function, while others are upregulated to delay or inhibit a target function, principally senescence. In addition, evidence suggests up-regulated mi-RNAs targeting STAT3 and JAK/STAT pathways in senescent cells, and their inhibition can alleviate age-related tissue dysfunction [194]. Nevertheless, the identification of the mechanism pathway might be hampered by the increasing number of the predicted targets outlined in the different mi-RNA databases. One can speculate that tissue specificity is primordial in modulating (promoting or inhibiting) synthesis, loading inside exosomes, and release of a mi-RNA or a panel of mi-RNAs whose combined action is pivotal for the target function. Further studies of multiple mi-RNA and their interaction in health, youth, and the elderly can make a real difference in our understanding of the mechanism of aging and aging-related diseases. For example, exosomes isolated from bone marrow stromal cells of young and aged C57BL/6 mice showed a similar size distribution and concentration. Mi-RNAs were differently enriched, and the amount of mi-RNA-183-5p was increased in aged cells [178]. Preclinical studies are actually developing mi-RNA mimics and mi-RNA inhibitors using exosomes as therapeutic drug delivery in vivo [195]. Interestingly, adipose tissue-derived exosomes from rats and pork have shown that xenogeneic injection of exosomes between these two species induced tissue repair and regeneration proving then their low immunogenicity [196].

Importantly, the results reported with exosomes highlight the potential of broad overlapping observations suggesting the need for shrewdly studying design through the identification of the parent cell, the active mi-RNA to be packaged in the exosome cargo and, the interaction with the specific target. For instance, exosomes were used in pre-clinical and clinical reports and induced anti-inflammatory responses as an indirect functionality or a side-effect. On the other side, variations in the culture conditions of parent cells (somatic cells, MSCs) and purification techniques lead to different inconsistencies in comparing the pools of the reported pre-clinical results. Therefore, we have already stated the need to standardize the operating procedures of MSCs. Similar approaches are required to prevent any changes in exosome cargo composition and downstream functional characteristics. Studies should be deeply investigated probably by using the considerable insights from the growing clinical trials conducted with exosomes in the treatment of severe acute respiratory syndrome coronavirus 2 (SARS-CoV-2) or the experienced MSC-based drug development.

Robust purification and characterization have been performed in designing release criteria of exosomes. More than 200 pre-clinical studies have used MSC-derived exosomes, mostly in cardiovascular, neurological, and kidney diseases [197]. Exosomes can be isolated from several sources: cultured cells, cell culture conditioned media, tissues, and biological fluids [1]. On one side, the purity of the exosome’s liberation depends on criteria such as (i) the state of the cell (in stress, stimulated, cryopreserved cell); (ii) its generation rate (exosome cargo and shape are affected in rapidly dividing cells); (iii) its nature (stem cells, tissue cells, cancer cells, etc.). On the other side, the used protocol for purification plays a crucial role in the exosome purity. For example, the gold standard for density-based purification is density gradient ultracentrifugation. Based on the size, the size-exclusion chromatography and tangential flow filtration are the most used. Based on exosome surface and immune affinity, CD63 and CD9 are the most used for their characterization [1,198,199] (Figure 2).

Recently, developed technologies like the three-dimensional nanostructured microfluidic chip, using arrays of micropillars, were used to capture exosomes with high efficiency through a combination of a specific recognition molecule [18]. Therefore, it is crucial to optimize the purification for high recovery and high reproducibility of exosomes for further manipulation.

Otherwise, cargo enrichment of exosomes is the most critical step in their manipulation, and numerous approaches have been mentioned. Exosome lipid membranes can be temporarily disrupted by physical or chemical treatment, sonication, and freeze-thaw cycles [200,201,202]. The fact that exosomes are negatively charged, manipulating the surface charge using magnetic and electric fields will lead to the encapsulation of the molecules [203]. These approaches are non-specific and can negatively affect the integrity of the membrane leading to a low exosome recovery.

In the same context, several studies used the ALIX protein from the ESCRT-dependent pathway to enrich exosomes with mi-RNAs [204]. Another passive cargo loading method consists simply of overexpressing our target in the media of parental cells until the uptake [198]. However, enriching exosomes by using gene editing and targeting ALIX in combination with other ESCRT seems to be more precise and specific.

Exosome release has not been fully evaluated. Although, different approaches were carried out to stimulate their secretion, such as the use of SNARE proteins [62], Rab27, Rab11, and Rab35 [73], ionophores like calcium signaling [73], and ARF6 [205]. However, intracellular pathways of the parent cells could be affected, impacting all the processes of exosome secretion and cargo enrichment.

Once these criteria are optimized and standardized, the final exosome should also fulfill the GMP requirements for therapeutic use, providing that functional assays are confluent successful. In aging-related diseases, the therapeutic properties of exosomes are promising and have opened the way to free-cell regenerative medicine. Nevertheless, little is reported on the underlying involvement of exosome-derived mi-RNA on the cellular and molecular mechanisms of aging.

It is intriguing to say that one mi-RNA has different pathways, and one can ask how this mi-RNA discerns which target and which pathway to activate; (i) is it depending on the tissue or the organ; (ii) the expression or expression level of their mRNA target and their niche; or (iii) all the interactions and mechanisms underlying the disease or the disorder. The aged individuals in the global population are increasing, adding a huge weight to the health care system. Comprehensive knowledge is still needed for more understandings of the relationships between mi-RNAs in the aging process and the improvement of exosomes’ cargo enrichment and delivery as drug vehicles.

## Figures and Tables

**Figure 1 biomedicines-09-00968-f001:**
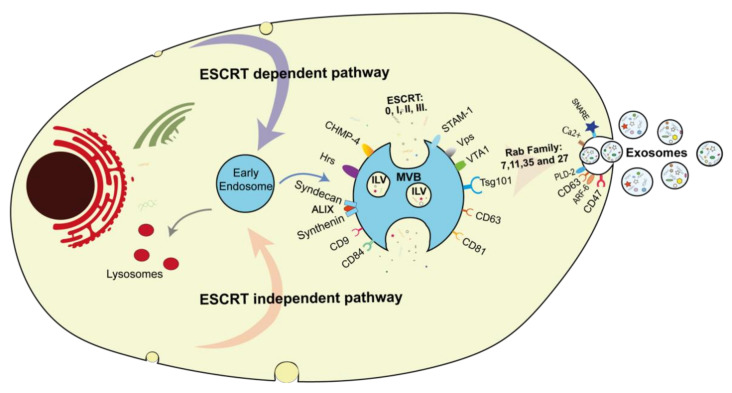
The biogenesis of exosomes, starting from the early endosomes, including both endosomal-sorting complex required for transport pathway (ESCRT). In the ESCRT pathway family (in the upper half of the cell), four sub-proteins 0, I, II, III, and charged multivesicular body protein- 4 (CHMP-4), hepatocyte growth factor-regulated tyrosine kinase substrate (Hrs), Programmed cell death 6-interacting protein (ALIX), signal-transducing adapter molecule 1 (STAM-1), vacuolar protein sorting proteins (Vps), and tumor susceptibility gene 101 (*TSG-101*) are identified. The ESCRT independent pathway (in the lower half of the cell) includes tetraspanin protein (CD9, CD84, CD81, CD63, CD47) and syndecan-Syntenin-ALIX. Exosome release results from the interaction of the Rab family, including Rab-7, 11, 35, 27 with Tspan and the Soluble N-ethylmaleimide-sensitive-fusion attachment protein receptor (SNARE).

**Figure 2 biomedicines-09-00968-f002:**
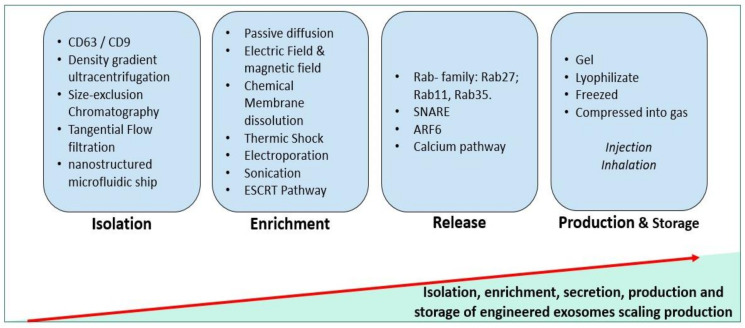
Steps for therapeutic exosome scale-up.

**Table 1 biomedicines-09-00968-t001:** Mi-RNA interaction in organs in the aging process. IRP: Iron responsive proteins; Wnt-3a: Wingless-type MMTV integration site family member-3A; p85α: Phosphatidylinositol 3-kinase regulatory subunit alpha; IGF-1: insulin-like growth factor I; B-myb: Myb-related protein B; DNMT: DNA methyltransferase; TGF-β1: transforming growth factor beta 1; Cx43: Connexin 43; RANKL: receptor activator of NFκB ligand; HMGB1: high-mobility group box-1; PTEN: phosphatase and tensin homolog; Akt: protein kinase B; HaCaT: spontaneously immortalized human skin keratinocytes; TIMP-3: tissue inhibitor of metalloproteinase 3; TIAM-1: T-lymphoma invasion and metastasis-inducing protein 1; Rho-B: RAS homolog family member B; MeCP2: methyl CpG binding protein 2; XIAP: X-linked inhibitor of apoptosis; ECM: extracellular matrix; COL16A-1: Collagen XVI; DEJ: dermal–epidermal junction; TNF-α: tumor necrosis factor-α; SIRT: Sirtuins; mTOR: mechanistic target of rapamycin; BMP2: bone morphogenetic proteins 2; Hmox1: heme oxygenase-1; P38-MAPK: p38 mitogen-activated protein kinases; c-Jun: N-terminal protein kinase; IP3: Inositol 1,4,5-trisphosphate; NPC: neural precursor cells proliferation; NA: not available; hUCB-MSC: human umbilical cord blood-derived mesenchymal stem cells; Foxg-1: forkhead box G1.

mi-RNA	Organ	Target	Interactions	Reference
mi-RNA-29	Brain	IRP	- mi-RNA-29 upregulation appears to be conserved as an aging signature and a common mechanism across different tissues. Neuronal-specific mi-RNA-29 loss of function induces accelerated expression of aging phenotypes. - Directly implicated between lifespan and reproduction.	[149,168]
	Muscle	Wnt-3a	- Aging-induced muscle senescence via Wnt-3a leading to the suppression of several signaling proteins (p85α, IGF-1, and B-myb) that act coordinately to impair the proliferation and contributing to muscle atrophy.	[159]
	Heart	DNMT TGF-β1	- mi-RNA-29 down-modulation is responsible for the accumulation of collagens and 5′methyl-cytosine.	[169,170]
mi-RNA-21	Bone	- RANKL - HMGB1	- The reduced level of Cx43 in osteocytes in old mice leads to apoptosis. - The release of RANKL and HMGB1 induce osteoclast resorption. - Gap junction communications by Cx43 channels maintain osteocyte viability via downstream regulation of mi-RNA-21, leading to the subsequent inhibition of PTEN activity and preservation of the Akt survival pathway.	[171]
	Skin	TGF-β1	- Promotes keratinocyte migration and boosts re-epithelialization during skin wound healing in HaCaT cells. - mi-RNA-21 might promote keratinocyte migration by inhibiting the expression of TIMP-3 and TIAM-1.	[172]
	Endothelial cells	Rho-B	- mi-RNA-21 acts as a negative modulator of angiogenesis. Its over-expression reduced endothelial cell proliferation, migration and to form tubulogenesis through the repression of RhoB expression.	[173]
mi-RNA-181	Brain	MeCP2 XIAP	- Overexpression of mi-RNA-181c in cultured astrocytes results in increased cell death when exposed to lipopolysaccharide as a model of inflammation. - mi-RNA-181 expression is regulated by inflammatory stimuli and modifies the proliferation of astrocytes and their sensitivity to death in an experimental model of neuroinflammation.	[174]
	Skin	COL16A-1	- Collagen XVI is a minor component of the skin ECM and is expressed in the DEJ zone of the papillary dermis connecting ECM proteins to cells, ensuring mechanical anchorage of the cell and outside-inside signal transduction. - mi-RNA-181a is increased during the human dermal fibroblast’s senescence, and its overexpression is sufficient to induce cellular senescence in early-passage cells.	[175]
mi-RNA-34	Heart	TNF-α	- mi-RNA-34a can inhibit autophagy after ischemia-reperfusion by targeting TNF-α and reduce myocardial injury (playing a protective role by reducing cell death, functional decline).	[176]
	Bone	Sirt1	- mi-RNA-34a expression is significantly increased with age in mice. - Induce senescence and decrease Sirt1 expression of primary bone marrow cells.	[147]
	Brain	SIRT1/mTOR	- Upregulated in aged mice, its downregulation has a protective role against aging-associated cognitive impairment.	[148]
mi-RNA-204 & mi-RNA-211	Heart	BMP2	- Attenuate vascular calcification and aging in a paracrine manner through an exosomal mi-RNA-204/ mi-RNA-211.	[177]
mi-RNA-183-5p	Bone	Hmox1	- Transfection of bone marrow stromal cells with mi-RNA-183-5p mimic reduces cell proliferation and osteogenic differentiation, increases senescence, and decreases protein levels of Hmox1.	[178]
mi-RNA-570	Epithelial cells	p38 MAP kinase-c-Jun	- Inhibiting mi-RNA-570-3p rejuvenates cells via restoration of sirtuin-1, reducing many of the abnormalities associated with cellular senescence.	[158]
mi-RNA-135-a-5p	Brain	IP3 pathway	- mi-RNA-135a inhibition stimulates NPC leading to increased neurogenesis, but not astrogliogenesis, and re-activates NPC proliferation in aged mice.	[179]
mi-RNA-127 -3p	Skin	NA	- mi-RNA-127-3p is an epigenetic activator regulating the transition from repair to remodeling during skin wound healing and may also induce age-related defects, pathological scarring, and fibrosis, all linked to myofibroblast senescence.	[180]
mi-RNA-124	Skin	NA	- mi-RNA-124 increases skin senescence and decrease during tumorigenesis.	[181]
	Liver	Foxg-1	- hUCB-MSC derived exosomal mi-RNA-124 could promote rat liver regeneration partial hepatectomy via downregulation of Foxg-1.	[106]

## Data Availability

Not applicable.

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
