# Peer review of "Exosomes and Micro-RNAs in Aging Process"

_biomedicines, 2021, doi:10.3390/biomedicines9080968_

Round 1
Reviewer 1 Report
The purpose of this work is to gain information about exosomes and their implication in aging process. with the emphasis on the role of exosome-derived microRNAs. The work contains an extensive chapter on the biogenesis of the exosomes, their cargo, and release. Then, various mechanisms influencing the aging process are discussed in detail. Finally, the last two chapters concern the role of exosomes and microRNAs in this process. The topic is very interesting, up to date and valuable however the goal was not fully achieved. Besides, the content of the manuscript needs to be put in order.
Contrary to the announcement in the title there is only a one chapter (6) devoted to role of microRNAs in aging process. The title do not reflect the whole content of this manuscript and should be modified and fitted more accurately to the included information. e.g. Exosomes and microRNAs in aging process.
On the other hand, however, the manuscript does not cover all the key information related to the implication of the exosomes in aging. Authors did not discuss the effects of different types components transmitted by exosomes on the aging process. Next to microRNAs also other biomolecules like mRNAs (including targets for microRNAs), proteins and lipids etc. are also transported in exosomes. They are even mentioned by the authors but not discussed. An additional information about role of exosomes and their cargo in aging would significantly enrich this manuscript.
However, if the authors prefer to focus on microRNAs only, they should significantly expand the Chapter 6 and pay attention on other small RNA molecules related to aging and associated with exosomes as well as microRNA-regulated genes contributed to senescence. For example please see doi: 10.3389/fnagi.2019.00232 or doi: 10.1016/j.addr.2012.07.010
The work requires a thorough English language check. Below are some examples of errors. The authors did not introduce line numbering, therefore it is difficult to refer to specific phrases. However, I will try to point out a few examples:
Page 5
single strand tails – shouldn’t be talis
a ribonucleoprote ???
The whole sentence “Cellular aging is due to… is not clear
Page 9
Exosomes derived stem-cell - ???
The whole sentence “Inflammatory cytokines … and the next one are not clear
more exosomes cell with different compositions
Page 10
miRNA derived exosomes
Another – should be another
Author Response
We want to thank very much the reviewer for his efforts and comments.
- The manuscript has beencompletely rearranged. Many paragraphes were moved and enriched in additional appropriate references related to the field and rewritten for more scientific understanding. We agree that some gaps remained within the text and required modifications;
- the title have been changed as suggested by the reviewer;
- As we have choosen to focus on the role of mi-RNAs on the aging process, a panel of new references has been added to enrich and highlight their involvement in regulating the mechansim pathways of longevity;
- the references suggested by the reviewer and related to senescence have been added to the text;
- grammatical and spelling errors have been corrected;
- English has been corrected by a native english person.
Reviewer 2 Report
The manuscript is a perspective article to summarize and highlight the merits of systems based on exosome derived microRNAs and aging interactions. The article places emphasis on general molecular mechanisms related to aging dysfunction, exosomes implication in aging process and the importance of mi-RNA in aging function. The authors also discussed the challenges and future aspects of the exosome derived miRNA for aging-related disease. In overall, the perspective manuscript is very comprehensive and worth publishing after addressing the following concerns. 1. The resolution of figure should be improved. 2. In the introduction part, there is a lack of references (e.g., ACS applied materials & interfaces 9 (33), 27441-27452; Nano Today 37, 101066; Advanced Functional Materials 28 (18), 1707360) on the overview of exosome purification approaches. 3. The “the importance of mi-RNA in aging function” should only focused on “exosome derived mi-RNA in aging function”. other exosomes unrelated work should be moved to “general molecular mechanisms part”. 4. In case of exosomes derived microRNA employed as therapeutic to treat aging related diseases, more discussions and detailed descriptions of referenced examples (especially add figure to represent these interesting examples) in the section is needed. 5. The authors discussed the current challenges of the field. However, it will be more useful for the reader if the authors expand the discussions about the possible ways to address these current challenges.Author Response
Dear reviewer,
thank you very much for reviewing this manuscript and for your kind evaluations and comments to which we have tried to give appropriate answers.
- the colour and resolution of the figure 1 were improved;
- the introduction section was rearranged and the suggested references included to emphasis the technical procedures of exosome purification;
- paragraphe 6 only reports the mi-RNA associated mechanisms of aging, meanwhile, the different molecular pathways modulating aging are summarized and moved to the paragraph 5;
- and 5. Actually, aging related diseases are largely addressed according to exosomes derived mi-RNA involvement in different scientific reports. Moreover, mi-RNA signature and expression levels is likely used as biomarkers for therapeutics in both diagnosis and disease progression. In this manuscript, we decided to more focus on the involvement of mi-RNAs in the process of tissue longevity and several findings have been cited to emphasis their role as promoting or inhibiting cell senescence and regeneration in young and aged individuals. Also, some strategies using exosomes derived mi-RNAs or exosomes as therapeutic drug delivery were suggested and discussed in the different steps of exosome enrichment and production to meet the current GMP requirements. The figure 2 indicates these perspectives in therapeutic.
Reviewer 3 Report
The review submitted by Hamdan and collegues addresses an interesting topic: which is the interaction (if any) between micro-RNAs derived from exosomes and aging?
The authors performed an extensive revision of literature, but the outcome colud be improved. The data presented are not well integrated and a real connection between exosome derived miRNA anche aging is not clear. Moreover english language needs extensive editind since often it is hard to understand the text.
Figure 1 needs to be improved: the color choice makes it hard to read.
Please check reference n.58
It is not clear which is the difference between genome instability (4.1) and DNA damage (4.2).
A more extentesive discussion could help in understanding the interaction claimed in the title
Author Response
- English editing was performed by a native english person;
- the figure 1 was improved in colour and resolution;
- The reference 58 was corrected;
- to be more readable, genome instability and DNA damages were included in the same paragraph;
- the whole manuscript has been rewritten to emphasis the interractions between tissue functionality, cell regeneration and mi-RNAs. Exosomes derived mi-RNA targeting different tissues and involving different pathways were also reported in association to aging in young and adult individuals. The discussion was highlighted by describing new strategies in developing the field of exosomes derived mi-RNA for both aging and aging associated diseases. Exosome production meeting GMP requirements might be used in therapeutic perspectives as drug delivery or to enhance an aging specific mi-RNA secretion.
- the title of the manuscript has been changed as it focuses mostly on mi-RNAs involving in aging
- spelling errors were corrected
Round 2
Reviewer 1 Report
The corrections made are sufficient. The manuscript can be accepted for publication in its present form.